# Effects of organic-inorganic complex fertilizer on the growth and physiological characteristics of 'Qi-Nan' agarwood from *Aquilaria sinensis* (Lour.)

Jingyue Huang[1,2], Yunlin Fu[1], Zhu Yu[3], Xueting Li[1], Siyu Zheng[1], Fenyong Tang[3], Penglian Wei [1]*

1 Guangxi Collegesand Universities Key Laboratory for Cultivation and Utilization of Subtropical Forest Plantation, College of Forestry, GuangxiUniversity, Nanning 530004, China, 2 Guangxi Gaofeng State Owned Forest Farm, Nanning, China, 3 Guangxi Forest Inventory & Planning Institute, Nanning, China

* weipenglian@gxu.edu.cn

## Abstract

This paper is to study the effects of different levels of fertilization on the growth, physiological characteristic of 'Qi-Nan' agarwood seedlings. The experiment was conducted under potted conditions using a single factor experiment. Organic-inorganic complex fertilizer was used as the basal fertilizers, and 7 fertilizer gradients ranging from 0 g/plant to 23 g/plant were set up. The result showed that: (1) within a certain range, with the increase of organic-inorganic complex fertilizer application, the growth and physiological activity of 'Qi-Nan' had an incremental effect, but this effect would be inhibited when the fertilizer application was too high. (2) After fertilization, the contents of free proline (FP) and malondialdehyde (MDA) decreased, while the contents of osmotic regulatory substances and chlorophyll increased. (3) The correlation analysis showed that seedling height and ground diameter of 'Qi-Nan' seedlings were significantly positively correlated with soluble sugar content, peroxidase activity and chlorophyll content. (4) Membership function analysis showed that organic-inorganic complex fertilizer had a significant effect on the growth of 'Qi-Nan' seedlings, and the optimal fertilizer level of single application of organic-inorganic complex fertilizer was 7-11g/plant. Therefore, the optimal application level of organic-inorganic complex fertilizer for the seedling cultivation of 'Qi-Nan' was 7g/ plant.

## 1. Introduction

Agarwood (Chen-Xiang in Chinese), the resinous heartwood from the *Aquilaria* species (Thymelaeaceae) [1], has widely used in medicine, aromatherapy, perfume industry, religious culture, and other fields. It is a valuable traditional medicine and natural spice [2–4]. 'Qi-Nan', also known as Kanankoh, Kyara [5,6] or Chi-Nan [7]. The wild 'Qi-Nan' is usually considered to be the highest quality wild agarwood wood, and its price is thousands of times higher than ordinary agarwood [8]. However, due to the destruction of the natural environment and excessive deforestation caused by economic interests, wild *Aquilaria* species are endangered [9]. This genus is now listed in Appendix II of the Convention on Internal

**Data availability statement:** All relevant data are within the manuscript and its Supporting information files.

**Funding:** The Guangxi Forestry Science and Technology Project (Guilin Kezi [2021]9) and the Guangxi Forestry Science and Technology Project (Guilin Scientific Research [2022ZC] No. 91).

**Competing interests:** The authors have declared that no competing interests exist.

Trade in Endangered Species of Wild Fauna and Flora [10]. Additionally, the low yield of wild *Aquilaria* species cannot meet the market demand. Therefore, planting agarwood trees have become an important way to obtain agarwood. In recent years, grafted 'Qi-Nan' agarwood tree has emerged. It is a high-quality agarwood obtained by grafting and breeding from natural wild agarwood trees [11,12]. *Aquilaria sinensis* "Reke2" is an example of high-quality cultivated 'Qi-Nan' propagated by grafting with the ordinary germplasm of *Aquilaria sinensis* [13]. It has been cultivated in places such as Hainan, Guangdong, Guangxi, and Yunnan, with excellent results. Compared to trees grown from ordinary agarwood seedlings, grafted 'Qi-Nan' agarwood can more easily produce higher-quality agarwood and has the advantages of early incense-bearing and high yield [11,14]. In order to satisfy the demand of the agarwood market, 'Qi-Nan' agarwood is being vigorously promoted and planted in China. According to preliminary statistics, 5,766 acres of 'Qi-Nan' agarwood seedlings have been planted nationwide [15–17]. Fertilization is a key measure for seedling planting [18] and is essential for the healthy growth of trees, which is the basis for forming more agarwood.

China's traditional crops and forestry on the planting of chemical fertilizer used in a single species, organic fertilizer accounted for a low proportion of fertilizer nutrient distribution ratio unreasonable is an important reason for the low utilization of fertilizer, and over-fertilization will cause a series of environmental pollution problems [19]. Some studies have pointed out that under the same nutrient content conditions, the nutritional effect of organic fertilizers applied in combination with chemical fertilizers exceeds that of single application of chemical fertilizers or single application of organic fertilizers, and at the same time, it plays a role in promoting the mineralization of organic fertilizers, activating the phosphorus in the soil, reducing the fixation of inorganic phosphorus, and extending the fertilizer supplying performance of chemical nitrogen fertilizers [20]. The application of organic-inorganic complex fertilizer is mostly seen in crops [14, 21–23], and its application has reached a certain scale, and the related research has also reached a certain degree. However, the application of organic-inorganic fertilizers on forest trees is relatively small, only focusing on the application of common tree species, such as *Cinnamomum camphora* [24], *Larix gmelinii* [25], *Eucalyptus* [26], etc. The application of organic-inorganic complex fertilizer on agarwood is rarely researched, especially on such precious tree species as 'Qi-Nan' agarwood.

At the fertilization level of agarwood plants, some scholars have studied the fertilization methods of agarwood. For example, the combination application of 10 g biological bacterial fertilizer and 20 g complex fertilizer has the best effect on promoting the growth of *Aquilaria sinensis* seedlings and enhancing soil fertility [27]. The use of a combination of inorganic fertilizer, organic fertilizer and biofertilizer treatments can better promote the growth of agarwood seedlings [28]. In addition, the fertilization method of *Aquilaria malaccensis* was also preliminarily explored. The photosynthetic capacity, growth amount and biomass of *Aquilaria malaccensis* were maximized when N was applied at 3 000 mg/plant. And when N was applied at 3.637g/ plant, P was applied at 2.5g/ plant and K was applied at 1.25g/ plant, the maximum biomass of *Aquilaria malaccensis* could be obtained at 27.688 g [29]. At present, however, there is a lack of studies on the fertilization and growth of 'Qi-Nan' seedlings. For example, the effects of different fertilization level on the growth and physiology of grafted 'Qi-Nan' agarwood seedlings have not been reported. The background that the quality and yield of 'Qi-Nan' agarwood are far from meeting the market demand, how to adopt a reasonable fertilization mode to achieve efficient seedling breeding is a significant problem; this problem needs to be solved urgently in the current cultivation process of 'Qi-Nan' agarwood seedlings.

Therefore, in this study, organic-inorganic complex fertilizer was used as base fertilizers to study the effects on the growth and physiological characteristics of 'Qi-Nan' agarwood seedlings.

The purpose of this study was threefold: (1) to determine the changes of the growth and physiology of 'Qi-Nan' seedlings under different fertilizing amounts; (2) to determine which physiological and nutrient indexes were the key factors affecting the height and ground diameter of 'Qi-Nan' seedlings; (3) to explore the level of organic-inorganic complex fertilizer application that is most conducive to the growth and development of 'Qi-Nan' agarwood seedlings.

## 2. Materials and methods

### 2.1. Experimental materials

The experimental varieties were obtained from the seedling breeding base in Nalou Town, Nanning, Guangxi (108°63'E, 22°60'N), which has a humid subtropical monsoon climate, with an average annual rainfall of 1304.2 mm, an average annual temperature of 21.6 °C and an average annual relative humidity of 79% [30]. Grafted 'Qi-Nan' agarwood seedlings with healthy and consistent growth for 2 months were selected as experimental materials in October 2021. The average seedling height was 17.60 ± 0.33 cm (average ± standard deviation) and the average ground diameter was 4.95 ± 0.29 mm (average ± standard deviation). The roots of the seedlings were kept clean before transplanting and transplanted into a plastic pot (with a tray) with a diameter of 264 mm and a height of 239 mm. The local yellow loam and sandy soil were mixed 1:1 as potting soil, and each pot was filled with 0.5 kg soil. The background values of the soil were measured before the experiment. The pH was 4.07, the alkali-hydrolyzed nitrogen was 29.43 mg/kg, the available potassium was 77.69 mg/kg, the available phosphorus was 30.14 mg/kg, and the organic matter content was 17.1 g/kg. The fertilizer used in the test was potassium sulfate compound fertilizer, produced by Miaoran Agricultural Chemical Technology Co., LTD., with total nitrogen ≥ 15%, total potassium ≥ 15%, total phosphorus ≥ 15%, and total nutrient content ≥ 45%.

### 2.2. Experiment design

The experiment was a single factor experiment of organic-inorganic complex fertilizer. 7 levels of organic-inorganic complex fertilizer were set to fertilize the current year's sown seedlings of 'Qi-Nan' agarwood. A total of 7 treatments were set up, CK (0 g/ plant), 1 (3 g/ plant), 2 (7 g/ plant), 3 (11 g/ plant), 4 (15 g/ plant), 5 (19 g/ plant), 6 (23 g/ plant). Each treatment was 12 plants with 3 replications. After a month of rejuvenation period, the grafted 'Qi-Nan' agarwood seedlings were fertilized. The background values of seedling height and ground diameter were measured before fertilization. All fertilization treatments were completely consistent in the management of supply, weeding, pest control and so on during the experiment, to minimize the experimental errors in the process of fertilization treatment. Fertilizer was applied twice, in November 2021 and May 2022, respectively, because of the seasonal nutrient uptake pattern of seedlings considered in the pot experiment. Evenly sprinkle the corresponding amount of organic and inorganic complex fertilizer on the surface of the potting soil, and then water when fertilizing. The experiment was conducted from November 2021 to November 2022, and the experimental period was one year.

### 2.3. Sample collection and determination

**2.3.1. Determination of growth index.** The plant height and ground diameter of 'Qi-Nan' agarwood seedlings were measured with steel tape (accuracy of 0.1 cm) and vernier caliper (accuracy of 0.1mm), respectively. The measurements were taken from November 2021 to November 2022, and the heights and ground diameters of all potted seedlings were measured every other month for a total of 12 times, each at the exact location.

**2.3.2. Measurement of physiological indexes and leaf nutrient content.** For each treatment, three well-grown seedlings were randomly selected, and the upper branches were used to count the 2nd to 5th mature functional leaves from the terminal bud downwards. These leaves were wiped clean, put into plastic bags, labeled, and temporarily stored in ice boxes; after collection, they were brought back to the laboratory and stored in an ultra-low-temperature refrigerator (-70 °C). The contents of chlorophyll, soluble sugar, soluble protein, free proline, and malondialdehyde were determined by the acetone extraction method [31], anthrone colorimetric method [32], Coomassie brilliant blue method [33], acid ninhydrin colorimetric method [34], and thiobarbituric acid colorimetric method [32], respectively. The activities of superoxide dismutase and peroxidase were determined by acid ninhydrin colorimetry [35] and guaiacol colorimetry [32]. The measurement time was February 2022, May 2022, August 2022, and November 2022, with a total of 4 measurements.

**2.3.3. Data analysis and processing methods.** The data were organized by Excel 2016; The SPSS 25.0 software (SPSS Inc., Chicago, IL, USA) was used to perform the statistical analyses by ANOVA and significance tests. $p \leq 0.05$ was considered to indicate statistically significant differences. Plotting was performed by origin 2021; The membership function method of fuzzy mathematics was used for the comprehensive evaluation of the growth and physiological indexes of 'Qi-Nan' agarwood under the effect of organic-inorganic complex fertilizer.

# 3. Results

## 3.1. Effects of organic-inorganic complex fertilizer on growth indexes of 'Qi-Nan' agarwood seedlings

**3.1.1. Effects of organic-inorganic complex fertilizer on seedling height of 'Qi-Nan' agarwood seedlings.** From Fig 1 that under different fertilization treatments, the dynamic change of plant height growth of 'Qi-Nan' agarwood seedlings showed an upward trend, and the growth rhythm was characterized by "slow-fast-slow" and showed an S-shaped curve. After May of the following year, the seedlings entered the rapid growth period, the seedling height growth increased significantly, in which the height growth of the seedlings was extremely rapid from June to August, and they entered the peak period of seedling height growth.

From Table 1, the differences in seedling height between groups of 'Qi-Nan' agarwood seedlings at the end of fertilization were significant (P < 0.05). The results of multivariate

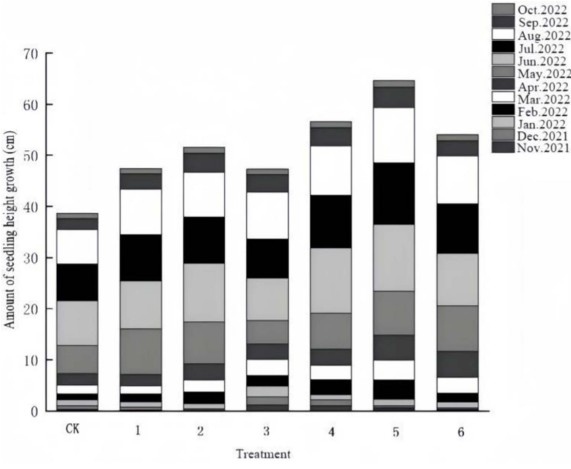

**Fig 1. Variation in seedling height growth of 'Qi-Nan' agarwood seedlings.**

**Table 1. Dynamics of seedling height growth of 'Qi-Nan' agarwood seedlings under different fertilization rates (Units: cm).**

| Month | Treatment | | | | | | | ANOVA |
|---|---|---|---|---|---|---|---|---|
| | CK | 1 | 2 | 3 | 4 | 5 | 6 | |
| Nov. | 17.9 | 17.2 | 17.1 | 17.5 | 17.8 | 18 | 17.7 | >0.05 |
| Dec. | 18.2 | 17.6 | 17.7 | 18.5 | 18.9 | 18.1 | 17.9 | <0.05 |
| Jan. | 18.9 | 17.9 | 18.1 | 19.6 | 20.5 | 18.4 | 18.4 | <0.05 |
| Feb. | 20 | 18.9 | 19.4 | 20.7 | 22.6 | 19.4 | 19.5 | <0.001 |
| Mar. | 21.3 | 20.7 | 23.1 | 23.6 | 24.6 | 21.7 | 21.1 | <0.001 |
| Apr. | 22.9 | 23.8 | 27.1 | 26.4 | 27.9 | 24 | 22.6 | <0.001 |
| May. | 25.2 | 28.9 | 32 | 29.6 | 30.9 | 27.3 | 24.9 | <0.001 |
| Jun. | 30.7 | 37.8 | 40.6 | 36.6 | 35.4 | 35.4 | 33.8 | <0.001 |
| Jul. | 39.5 | 46.3 | 53.7 | 49.4 | 43.7 | 46.9 | 43.2 | <0.001 |
| Aug. | 46.7 | 56 | 65.7 | 59.7 | 51.4 | 56 | 52.2 | <0.001 |
| Sep. | 53.5 | 65.4 | 76.5 | 69.3 | 60.6 | 64.7 | 61.1 | <0.001 |
| Oct. | 55.6 | 68.4 | 80.6 | 72.9 | 64 | 68.4 | 64.1 | <0.001 |
| Nov. | 56.6 | 69.6 | 81.8 | 74.1 | 65.1 | 69.5 | 65.2 | <0.001 |
| Total increment | 38.7±3.2De | 52.4±4.9Cbc | 64.7±5.6Aa | 56.6±3.2Bb | 47.3±4.2Ccd | 51.5±5.0Cbd | 47.5±3.8Ccd | <0.001 |
| Relative increment | 100.00% | 135.40% | 167.30% | 146.40% | 122.50% | 133.30% | 122.70% | <0.001 |
| P | 0.00 | 0.00 | 0.00 | 0.00 | 0.00 | 0.00 | 0.00 | |

Note: The data in the table are mean ± standard deviation; Different capital letters indicate highly significant differences between the same index, the same period, and different treatments at the 0.01 level (P < 0.01), and different lowercase letters indicate significant differences between the same index, the same period, and different treatments at the 0.05 level (P < 0.05).

analysis of variance showed that different amounts of organic-inorganic complex fertilizer treatment and different growth months had significant effects on plant height growth (P < 0.01). Among them, the seedling height and total growth of 'Qi-Nan' agarwood seedlings were the largest in treat 2, 81.8 cm and 64.7 cm, respectively, which were 144.5% and 167.3% of CK, followed by treat 3. The seedling height and total growth of 'Qi-Nan' agarwood in treat 4 were the smallest, 65.1 cm and 47.3 cm, respectively, which were 115.0% and 122.5% of CK.

Multiple comparisons of treatment can be seen from Table 2. As the concentration of organic-inorganic complex fertilizer increases, it is helpful for the height growth of 'Qi-Nan' agarwood. The seedling height growth of treatment 2 is the largest, followed by treatment 3, and then the seedling height growth gradually decreases with the increase of fertilization concentration. The concentration of organic-inorganic complex fertilizer that can promote the seedling height ranges from 7-11g/ plant.

The multiple comparisons of the main effect of months in Table 3 show that the monthly growth of seedling height first increases and then decreases with the fertilization time. Among

**Table 2. Multiple-factor comparison of treatments main effects on plant height growth of month.**

| Treatment | CK | 1 | 2 | 3 | 4 | 5 | 6 |
|---|---|---|---|---|---|---|---|
| PLH | 3.22Dd | 4.36BCbc | 5.39Aa | 4.72Bb | 3.94Cc | 4.29BCbc | 3.96Cc |

**Table 3. Multiple comparison of monthly main effect on plant height growth of 'Qi-Nan'.**

| Treatment | Feb. | May. | Aug. | Nov. |
|---|---|---|---|---|
| PLH | 0.84Dd | 2.82Cc | 8.90Aa | 4.51Bb |

them, the differences between each month are extremely significant, and the growth of seedling height is the largest in August.

In conclusion, different concentrations of organic and inorganic compound fertilizer have significant differences on seedling height, and the optimal concentration of organic and inorganic compound fertilizer topdressing for seedling height growth of 'Qi-Nan' agarwood is about 7-11g/ plant.

**3.1.2. Effect of organic-inorganic complex fertilizer on the ground diameter of 'Qi-Nan' agarwood seedlings.** From Fig 2 that in the early stage of fertilization (2021.11-2022.2), the effect of each treatment on the growth of 'Qi-Nan' incense was not obvious, and the growth of seedling diameter was still slow in March and April. From May to June is the peak period of the growth of the ground diameter, and the growth of the ground diameter in all treatments is larger in June. After June, the growth rate decreased, and the increase in diameter of each treatment was more average in different months from June to October.

From Table 4, after fertilization, the seedling height and ground diameter of 'Qi-Nan' agarwood seedlings had significant differences among groups (P < 0.05). The results of multivariate analysis of variance showed that different amounts of organic-inorganic complex fertilizer treatment and different growth months on plant height growth were highly significant (P < 0.01). Among them, treat 2 had the largest growth in diameter and total diameter, which were 160.38% and 214.24% of CK, followed by treat 3. Treat 6 had the smallest growth in diameter and total diameter, which were 0.83 mm and 0.33 mm less than that of the CK, and some of the seedlings in treat 6 were dead, which may be due to 'burnt' caused by the application of the high-concentration fertilizers.

From Table 5, multiple comparisons of the main effects indicated that with the increase in the topdressing amount of organic-inorganic compound fertilizers, the ground diameter growth of 'Qi-Nan' agarwood seedlings initially ascended and then declined. A high concentration of organic-inorganic compound fertilizers inhibited the growth of seedlings and even severely hampered the growth of seedling diameter. Among them, the seedling diameter growth was the highest under treatment 2, suggesting that a topdressing amount of 7g/plant was more conducive to the seedling diameter growth of 'Qi-Nan' agarwood.

From Table 6, the monthly growth of the ground diameter displayed a trend of initially rising and subsequently falling with the fertilization time. There was no significant difference

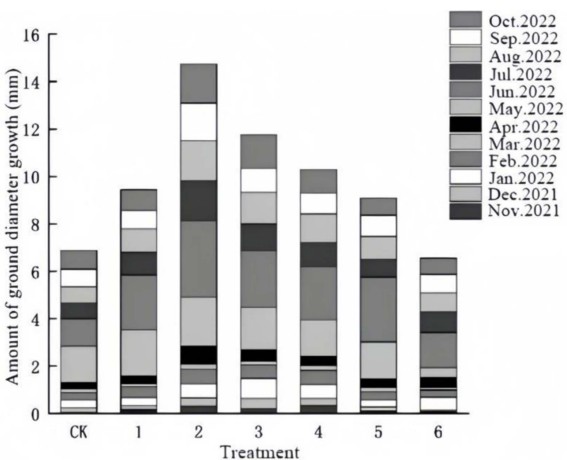

**Fig 2. Variation in ground diameter growth of 'Qi-Nan' agarwood seedling.**

**Table 4. Dynamics of ground diameter growth of 'Qi-Nan' agarwood seedlings under different fertilization rates (Units: mm).**

| Month | Treatment | | | | | | | ANOVA |
|---|---|---|---|---|---|---|---|---|
| | CK | 1 | 2 | 3 | 4 | 5 | 6 | |
| Nov. | 5.36 | 4.76 | 4.89 | 5.26 | 4.53 | 5.01 | 4.86 | <0.05 |
| Dec. | 5.42 | 4.92 | 5.2 | 5.47 | 4.87 | 5.13 | 4.93 | >0.05 |
| Jan. | 5.6 | 5.09 | 5.55 | 5.9 | 5.17 | 5.29 | 5.01 | <0.05 |
| Feb. | 5.94 | 5.44 | 6.15 | 6.74 | 5.75 | 5.59 | 5.55 | <0.001 |
| Mar. | 6.25 | 5.89 | 6.76 | 7.31 | 6.35 | 5.93 | 5.85 | <0.001 |
| Apr. | 6.4 | 6.02 | 6.99 | 7.48 | 6.55 | 6.11 | 5.97 | <0.001 |
| May. | 6.66 | 6.35 | 7.74 | 7.94 | 6.95 | 6.47 | 6.38 | <0.001 |
| Jun. | 8.2 | 8.28 | 9.79 | 9.74 | 8.48 | 8.03 | 6.8 | <0.001 |
| Jul. | 9.36 | 10.6 | 13.03 | 12.13 | 10.73 | 10.76 | 8.28 | <0.001 |
| Aug. | 10.02 | 11.56 | 14.71 | 13.26 | 11.74 | 11.52 | 9.16 | <0.001 |
| Sep. | 10.71 | 12.55 | 16.4 | 14.59 | 12.95 | 12.48 | 9.95 | <0.001 |
| Oct. | 11.45 | 13.32 | 17.98 | 15.61 | 13.82 | 13.37 | 10.73 | <0.001 |
| Nov. | 12.24 | 14.2 | 19.63 | 17.02 | 14.81 | 14.1 | 11.41 | <0.001 |
| Total increment | 6.88±0.89d | 9.44±1.01c | 14.74±1.50a | 11.76±1.24b | 10.28±1.10bc | 9.09±0.76c | 6.55±0.75d | <0.001 |
| Relative increment | 100.00% | 137.21% | 214.24% | 170.93% | 149.42% | 132.12% | 95.20% | <0.001 |
| P | 0.00 | 0.00 | 0.00 | 0.00 | 0.00 | 0.00 | 0.00 | |

Note: The data in the table are mean ± standard deviation; Different capital letters indicate highly significant differences between the same index, the same period, and different treatments at the 0.01 level (P < 0.01), and different lowercase letters indicate significant differences between the same index, the same period, and different treatments at the 0.05 level (P < 0.05).

**Table 5. Multiple-factor comparison of treatments main effects on diameter growth of month.**

| Treatment | CK | 1 | 2 | 3 | 4 | 5 | 6 |
|---|---|---|---|---|---|---|---|
| Grd | 0.58Ccd | 0.78BCbc | 1.22Aa | 0.98ABb | 0.8BC6b | 0.76BCbcd | 0.55 Cd |

**Table 6. Multiple comparison of monthly main effect on diameter growth of 'Qi-Nan'.**

| Treatment | Feb. | May. | Aug. | Nov. |
|---|---|---|---|---|
| Grd | 0.31Cc | 0.36Cc | 1.59Aa | 1.02Bb |

between February and May, yet a highly significant difference occurred between August and November. The disparity between August and November was also highly significant. These results imply that the growth of the ground diameter accelerates in August, then slows down and essentially stops in November.

In conclusion, the effects of different concentrations of organic and inorganic compound fertilizer on the soil diameter are significantly distinct, and the optimal concentration of organic and inorganic compound fertilizer for the soil diameter growth of 'Qi-Nan' agarwood seedlings is approximately 7g/plant.

### 3.2. Effects of organic-inorganic complex fertilizer on physiological and biochemical indexes of 'Qi-Nan' agarwood seedlings

**3.2.1. Effects of organic-inorganic complex fertilizer on the content of Osmotic mediating substances in 'Qi-Nan' agarwood seedlings.** According to the results of the effects on soluble protein (SP) (Fig 3a), the SP content of 'Qi-Nan' agarwood seedlings differed between groups (P < 0.05) in February, May, August and November. The results

of multivariate analysis of variance showed that the effects of different fertilizer amount of organic-inorganic complex fertilizer treatments and different growth months on SP content were highly significant (P < 0.01). Overall, the SP content of 'Qi-Nan' agarwood seedlings did not change significantly over time under different amounts of organic and inorganic compound fertilizer. With the increase of fertilizer application, the SP content generally increased first and then decreased. Except that the SP content of treat 2 in November was lower than that of treat 3, the SP content of treat 2 and treat 3 in other months was higher, indicating that the two concentration levels of fertilization had a better effect on the synthesis of SP content in seedlings. Within a certain range, the SP content of 'Qi-Nan' agarwood seedlings also increased with the increase of the amount of organic-inorganic complex fertilizer application, but the SP content would decrease beyond a certain concentration.

According to the results of the effect on soluble sugar (SS) (Fig 3b), the SS content of 'Qi-Nan' agarwood seedlings differed between groups (P < 0.05) in February, May, August and November. The results of multivariate analysis of variance showed that the effects of different amounts of organic-inorganic compound fertilizer treatment and different growth months on SS content were highly significant (P < 0.01). Overall, the SS content of 'Qi-Nan' agarwood seedlings showed an increasing trend over time, and the overall change was small. The SS content in February increased first and then decreased with the increase of fertilizer application. In August, the SS content of treat 3 and 4 was lower than that of CK, but entered November, the SS content showed an upward trend with the increase of fertilizer application, and the SS content of each fertilizer treatment was higher than that of CK, indicating that organic and inorganic compound fertilizer had a better promotion effect on the synthesis of SS. Meanwhile, the SS content of treat 6 reached the maximum.

**3.2.2. Effect of bio-organic fertilizer treatments on enzyme activity of 'Qi-Nan' agarwood seedlings.** From the results of the effect on peroxidase (POD) activity (Fig 4a), there were differences (P < 0.05) between groups of POD activity in 'Qi-Nan' seedlings in February, May, August and November. The results of multifactorial ANOVA showed that the

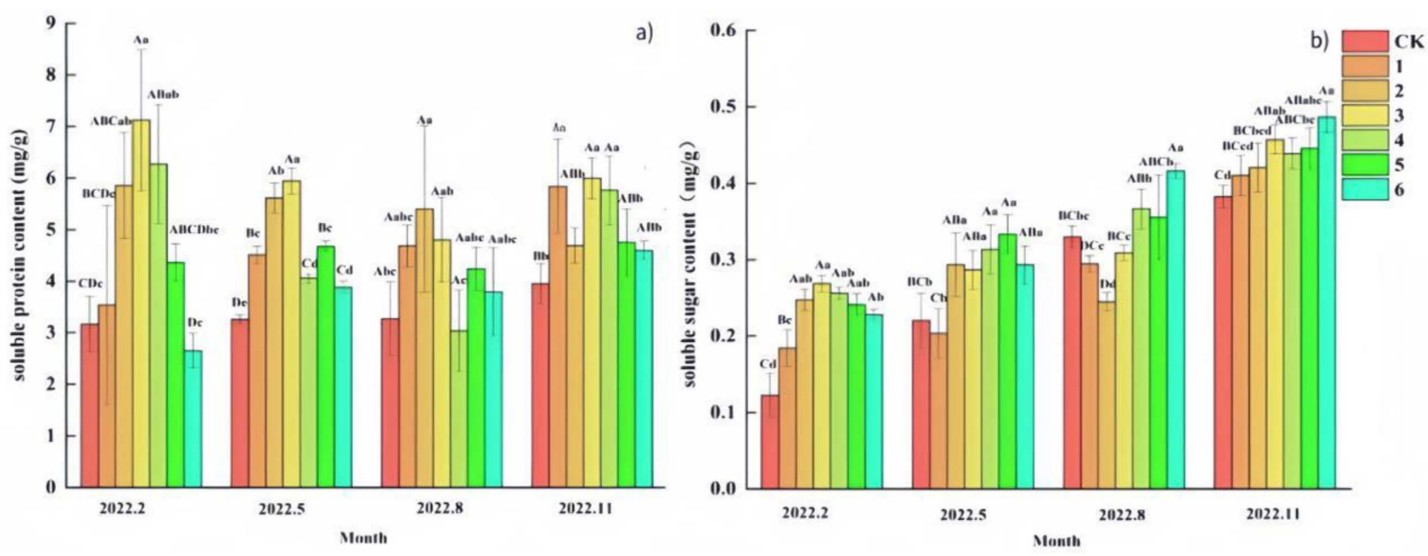

**Fig 3. Effects of different fertilizer treatments on soluble protein (SP) and soluble sugar (SS) content of 'Qi-Nan' agarwood seedlings.** (a) soluble protein; (b) soluble sugar. Different capital letters indicate highly significant differences between the same index, the same period, and different treatments at the 0.01 level (P < 0.01), and different lowercase letters indicate significant differences between the same index, the same period, and different treatments at the 0.05 level (P < 0.05).

effects of different fertilizer doses of organic and inorganic composite fertilizer treatments and different growth months on POD activity were highly significant (P < 0.01). Under different fertilization treatments of organic and inorganic compound fertilizers, the POD activity of 'Qi-Nan' seedlings showed an increasing and then decreasing trend with the increase of fertilization. Throughout the growth process, POD activity showed a general upward trend with time, indicating that peroxidase activity was weak in young seedlings and high in mature seedlings. POD activity was higher in all fertilization treatments than in CK. During the month of November, the POD activity of treatments 1, 2, and 3 increased compared to August, whereas treatments 4, 5, and 6 decreased compared to August, indicating that the POD activity continued to increase at low concentrations, and was higher in the early part of the year and lower in the later part of the year at high concentrations.

From the results of the effect on superoxide dismutase (SOD) activity (Fig 4b), there were differences (P < 0.05) between groups of SOD activity in 'Qi-Nan' seedlings in February, May, August and November. The results of multifactorial ANOVA showed that the effects of different fertilizer application rates of organic-inorganic composite fertilizer treatments and different growing months on SOD activity were highly significant (P < 0.01). The SOD activity of 'Qi-Nan' seedlings increased with the increase of organic and inorganic composite fertilizers in each fertilization treatment. In February and May, the SOD activity of treatment 6 was high compared to that of treatments CK, 1, 2, and 3 with low concentration, while in November it was low compared to that of treatments with low concentration, indicating that superoxide dismutase activity was higher in the early stage of the treatment with high fertilizer concentration, and declined or flat in the later stage of the treatment; the SOD activities of the control CK and the treatments 1, 2, and 3 with low concentration largely showed an increase with the increase of the concentration throughout the growth process, and the SOD activeness could be maintained with a sustained growth in the low concentration.

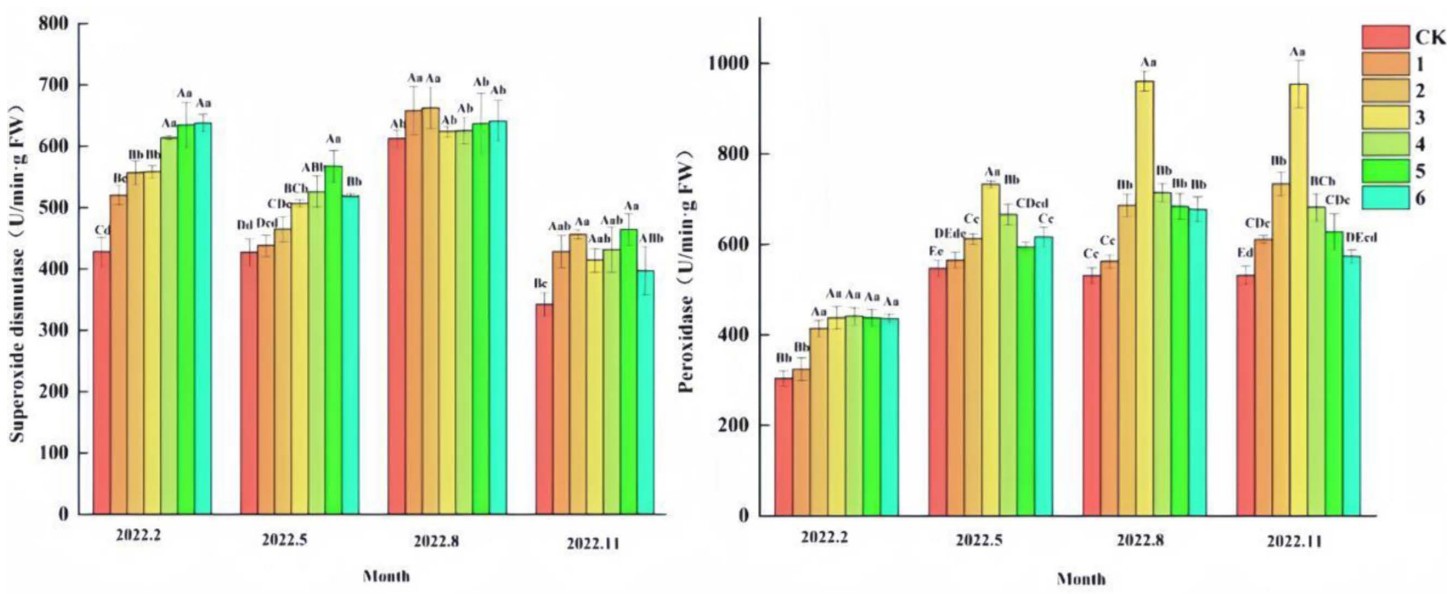

**Fig 4. Effects of different fertilizer treatments on peroxidase (POD) and superoxide dismutase (SOD) activities of 'Qi-Nan' agarwood seedlings.** (a) peroxidase; (b) superoxide dismutase. Different capital letters indicate highly significant differences between the same index, the same period, and different treatments at the 0.01 level (P < 0.01), and different lowercase letters indicate significant differences between the same index, the same period, and different treatments at the 0.05 level (P < 0.05).

**3.2.3. Effect of fertilization on malondialdehyde in 'Qi-Nan' seedlings.** From the results of the effect on malondialdehyde (MDA) (Fig 5), there were differences ($P < 0.05$) between groups in the MDA content of 'Qi-Nan' seedlings in February, May, August and November. The results of multifactorial ANOVA showed that the effects of different fertilizer application rates of organic-inorganic composite fertilizer treatments and different growing months on MDA content were highly significant ($P < 0.01$). The MDA content of 'Qi-Nan' seedlings varied greatly among different fertilizer treatments and different months throughout the growth process. The changes in MDA content of 'Qi-Nan' seedlings treated with different fertilization rates in different months were basically the same, generally showing a decreasing and then increasing trend. The MDA content of the treatments was comparable in February and November, both of which were significantly higher than in May and August, while the lowest MDA content was found in all treatments in August.

**3.2.4. Effects of fertilization on chlorophyll content of 'Qi-Nan' seedlings.** From the results of the effect on chlorophyll (Chl) content (Fig 6), the Chl content of 'Qi-Nan' seedlings differed between groups ($P < 0.05$) in February, May, August and November. The results of multifactorial ANOVA showed that the effects of different fertilizer rates of organic-inorganic composite fertilizer treatments and different growing months on Chl content were highly significant ($P < 0.01$). Overall, it appeared that the Chl content of 'Qi-Nan' seedlings increased and then decreased with the time of fertilization, reaching a maximum in August, when photosynthesis was strongest. Except for treatment 6, the Chl content of all treatments was higher than that of CK, indicating that organic and inorganic composite fertilizers can effectively promote the growth of Chl within a certain range of fertilizer application. The Chl content of treatment 6 was lower than that of CK in the periods of February, May, and August, and slightly higher than that of CK in November, indicating that the application of high concentration of organic and inorganic compound fertilizer in the early stage exerted a stressful effect on the 'Qi-Nan' sedum seedling, which inhibited Chl synthesis in the seedling. As the root system of the seedlings grew later in the season, the seedlings became more resilient, allowing the chlorophyll content to be gradually higher than that of CK.

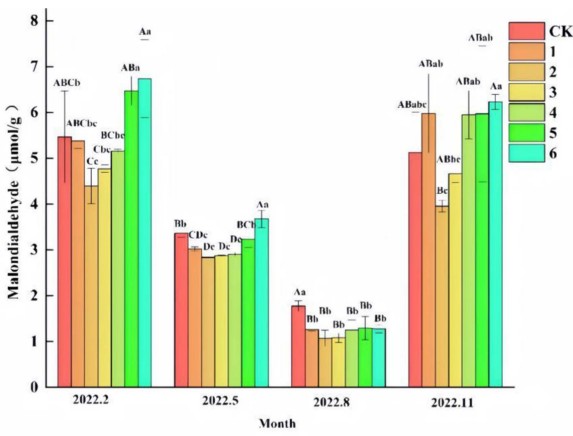

**Fig 5. Effects of different fertilizer treatments on the content of malondialdehyde (MDA) in 'Qi-Nan' agarwood seedlings.** Different capital letters indicate highly significant differences between the same index, the same period, and different treatments at the 0.01 level ($P < 0.01$), and different lowercase letters indicate significant differences between the same index, the same period, and different treatments at the 0.05 level ($P < 0.05$).

## 4. Comprehensive analysis of the indicators of 'Qi-Nan' seedlings

Correlation analysis of different growth physiological indexes in each month is shown in Fig 7. Seedling height was highly significantly positively correlated with SS and POD, and significantly positively correlated with Chl; Ground diameter was highly significantly and positively correlated with SS, Chl and POD; POD showed highly significant positive correlation with SS, significant positive correlation with SS, significant positive correlation with Chl and significant negative correlation with MDA; MDA showed highly significant negative correlation with Chl and SOD. This indicates that when the Chl content, SS content and POD activity

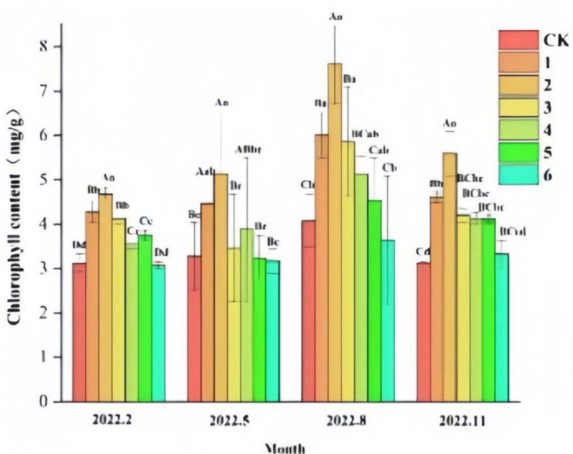

**Fig 6. Effects of different fertilization rates on chlorophyll(chl) content of 'Qi-Nan' agarwood seedlings.** Different capital letters indicate highly significant differences between the same index, the same period, and different treatments at the 0.01 level (P < 0.01), and different lowercase letters indicate significant differences between the same index, the same period, and different treatments at the 0.05 level (P < 0.05).

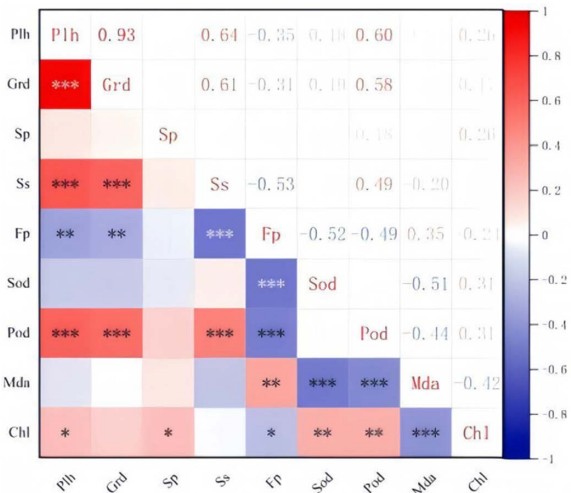

**Fig 7. Effects of different fertilization rates on chlorophyll content of agarwood seedlings.** (a) plant height (PlH); (b) ground diameter (GrD); (c) soluble protein (SP) (d) soluble sugar (SS); (e) Free proline (FP); (f) superoxide dismutase (SOD); (g) peroxidase (POD); (h) malondialdehyde (MDA); (i) chlorophyll (Chl). p<0.5: *; p<0.1: **; p<0.01: ***.

are higher, the seedling height and diameter of 'Qi-Nan' seedlings will grow better; on the contrary, when the Chl content, SS content and POD activity are lower and the MDA content is higher, the seedling height and diameter of 'Qi-Nan' seedlings will grow worse.

Based on the results of the above growth and physiological condition tests, membership function analysis was used to synthesize and analyze the results in Table 7. Based on the mean value of affiliation, it can be concluded that treatment 3 has the highest composite score, followed by treatment 2, treatment 5 and treatment 4. The composite ranking is 3 > 2 > 5 > 4 > 1 > 6> CK.

Among them, the score of treatment 5 is higher than that of treatment 4, which may be due to the fact that the malondialdehyde score of treatment 5 is much larger than that of treatment 4, while the scores of the other indicators do not differ much, which makes the mean value of the affiliation of each indicator of treatment 5 larger than that of treatment 4. It showed that treatment 5 was subjected to greater stress and produced more malondialdehyde than treatment 4; treatments 4 and 5 scored higher than treatments 1 and 6, indicating that treatment 6 was over-fertilized compared to treatments 4 and 5, and that treatment 1 was under-fertilized compared to treatments 4 and 5, and that over- or under-fertilizing was detrimental to the growth of seedlings; and treatment 1 scored higher than treatment 6, suggesting that organic and inorganic composite fertilizers at high concentrations inhibited the growth of seedlings.

## 5. Discussion

### 5.1. Effects of organic and inorganic compound fertilizer on growth indexes of agarwood seedlings of 'Qi-Nan'

Plant height, ground diameter and so on, as the key growth indicators of forest growth, can well reflect the growth state of forest [36]. The seedling height and diameter were higher than CK in all fertilizer treatment groups in this experiment, except for treatment 6, which had a slightly smaller diameter than CK. Among them, the seedling height and ground diameter of treatment 2 reached the maximum among the groups, which were 44.52% and 60.38% higher than CK, respectively. This suggests that moderate application of organic-inorganic complex fertilizers will promote the growth of 'Qi-Nan' seedlings, which is similar to the findings of olive tree [37], *Aquilaria sinensis* [28], *Aquilaria yunnanensis* [38] and *Corylus heterophylla* [39]. However, fertilization should be appropriate, too high or too low fertilization are not conducive to the growth of seedlings, too low fertilization cannot provide seedlings with

**Table 7. Analysis of seedling growth and membership degree of physiological effects of 'Qi-Nan' agarwood under different fertilization rates.**

| Index | CK | 1 | 2 | 3 | 4 | 5 | 6 |
|---|---|---|---|---|---|---|---|
| plant height | 0.00 | 0.51 | 1.00 | 0.69 | 0.34 | 0.51 | 0.34 |
| diameter | 0.10 | 0.34 | 1.00 | 0.68 | 0.41 | 0.33 | 0.00 |
| Soluble protein content | 0.00 | 1.00 | 0.39 | 1.08 | 0.96 | 0.42 | 0.34 |
| Soluble sugar content | 0.00 | 0.26 | 0.40 | 0.72 | 0.54 | 0.60 | 1.00 |
| Free proline content | 1.00 | 0.69 | 0.75 | 0.60 | 0.60 | 0.00 | 0.21 |
| Malondialdehyde content | 0.48 | 0.44 | 0.00 | 0.17 | 0.34 | 1.00 | 0.60 |
| chlorophyll content | 0.00 | 0.63 | 1.00 | 0.47 | 0.41 | 0.42 | 0.15 |
| Superoxide dismutase activity | 0.00 | 0.70 | 0.94 | 0.59 | 0.73 | 1.00 | 0.45 |
| peroxidase activity | 0.00 | 0.12 | 0.48 | 1.00 | 0.36 | 0.23 | 0.10 |
| Mean value of affiliation | 0.18 | 0.52 | 0.66 | 0.67 | 0.52 | 0.50 | 0.35 |
| Order of affiliation | 6 | 3 | 2 | 1 | 3 | 4 | 5 |

sufficient nutrients, while too high fertilization may cause seedlings of the "seedling burning" phenomenon, the seedlings have a toxic effect. In this experiment, under the treatment of 7-11g/plant of organic-inorganic complex fertilizer, the growth performance of 'Qi-Nan' seedlings were optimal, and the growth condition of the treatment group with more than 11g/plant became worse gradually with the increase of fertilizer application. This indicates that within a certain range of fertilizer application, a reasonable amount of fertilizer will significantly promote seedling growth, while too high or too low fertilizer application will be detrimental to seedling's growth. This is consistent with the results of the study of *Sapindus mukorossi* [40] and *Amorpha fruticose* [41] seedlings. With the increase of fertilizer application, the growth indicators of *Sapindus mukorossi* and *Amorpha fruticose* showed a tendency of increasing and then decreasing. In addition, the growth trend of seedlings in the whole stage is "slow, fast and slow", which may be attributed to the weak root system of seedlings in the early stage of growth, the application of organic-inorganic complex fertilizers and the cold weather in winter produced a certain degree of stress on the plant, which requires a period of slowing down the seedling period.

## 5.2. Effects of organic and inorganic fertilizers on the physiology of 'Qi-Nan' seedlings

The effects of different concentrations of organic-inorganic complex fertilizers on the physiological indexes of 'Qi-Nan' seedlings varied. In this experiment, the chlorophyll content, soluble protein content, and POD content of seedlings showed a trend of increasing and then decreasing with the increase in the amount of fertilizer, which is consistent with the results of different nitrogen, phosphorus, and potassium application on buckwheat leaves [42], and different concentrations of water-fertilizer coupling on greenhouse bag-grown tomatoes [43]. This suggests that a reasonable amount of fertilizer will improve the photosynthetic capacity of 'Qi-Nan' seedlings and promote plant carbon and nitrogen metabolism. SS and SP are both indispensable nutrients in plants, and they also play an important role in maintaining osmoregulation in plants [44]. When the concentration of organic-inorganic complex fertilizers was too high, the SS content kept increasing, indicating that the high concentration of organic-inorganic complex fertilizers promoted the SS content of 'Qi-Nan' seedlings better, while the content of SP appeared to be decreasing instead. The reason may be that the high concentration of fertilizer caused high protease activity in 'Qi-Nan' seedlings, which accelerated the rate of protein hydrolysis, while RNA transcription and translation were inhibited [45], resulting in a decrease in the SP content in 'Qi-Nan' seedlings. This is in line with the results emerging from the study of physiological and ecological adaptation of *Suaeda heteropptera Kitagawa* to salt stress alleviation [46]. FP and MDA are indicators of stress tolerance in plants [47], and with the increase of o organic-inorganic complex fertilizer application, the content of MDA and FP first decreased and then increased, indicating that the appropriate concentration of organic-inorganic complex fertilizer can reduce the environmental stress suffered by 'Qi-Nan' seedlings. Under adverse conditions, the metabolism of reactive oxygen species in plants is imbalanced with the scavenging of the defense system and the membrane structure is disrupted [48]. In contrast, POD and SOD, as protective enzyme systems in plants, are used to remove reactive oxygen species free radicals, relieve membrane lipid peroxidation of cells, and reduce the damage to plants caused by the hostile environment [49]. In the experiment, the difference of enzyme activity between different concentrations of organic-inorganic complex fertilizers was highly significant. From the point of view of the activity size, if the concentration was too low, the promotion of enzyme activity in 'Qi-Nan' seedlings was not obvious, and if the concentration was too high, the enzyme activity was inhibited by the fertilizers, so the

reasonable amount of organic-inorganic complex fertilizers could help to increase the enzyme activity in 'Qi-Nan' seedlings and enhance the seedling's resistance.

## 6. Conclusions

'Qi-Nan' agarwood is the highest quality and value agarwood on the market. However, no one has studied the suitable fertilization method for potted seedlings. Our study found that organic-inorganic complex fertilizer has a greater effect on the growth and development of 'Qi-Nan' seedlings, and that moderate amounts of organic-inorganic complex fertilizers will promote the growth of 'Qi-Nan' seedlings, while excessive amounts will have an inhibitory effect on their growth. Correlation analysis showed that the main factors affecting the seedlings' heights and diameters were POD, chlorophyll, and SS content, and all three of them had higher content around November, indicating that 'Qi-Nan' potted seedlings had strong photosynthetic capacity and better growth in November. Overall, the optimal fertilizer amount of 'Qi-Nan' agarwood potted seedlings is about 7-11g/ plant. However, the results obtained from the pot trials were limited and not sufficiently representative of long-term fertilization effects. Therefore, it is recommended to follow up with additional field trials of 'Qi-Nan' fertilization and continue the research so that the results of the trials will be of practical significance.

## Suporting information

**S1 File.**
(DOCX)

## Acknowledgments

The authors would like to thank Guangxi Miaoran Agricultural Chemical Technology Co., LTD for their supply of organic-inorganic complex fertilizers, thank the agarwood cultivation base in Nalou Town for providing the environment for pot experiment, and thank relevant staff for helping to cultivate the 'Qi-Nan' agarwood seedlings and measuring the data.

## Author contributions

**Conceptualization:** Penglian Wei.

**Funding acquisition:** Yunlin Fu.

**Investigation:** Jingyue Huang, Zhu Yu, Xueting Li, Siyu Zheng, Fenyong Tang.

**Software:** Jingyue Huang, Zhu Yu.

**Supervision:** Yunlin Fu.

**Writing – original draft:** Jingyue Huang, Zhu Yu.

**Writing – review & editing:** Penglian Wei.

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
