## [Decision Letter · Decision Letter 0]

22 Nov 2024

PONE-D-24-45249Effects of Organic-inorganic Complex Fertilizer on the Growth and Physiological Characteristics of 'Qi-Nan' Agarwood from Aquilaria sinensis (Lour.)PLOS ONE

Dear Dr. Penglian,

Thank you for submitting your manuscript to PLOS ONE. After careful consideration, we feel that it has merit but does not fully meet PLOS ONE’s publication criteria as it currently stands. Therefore, we invite you to submit a revised version of the manuscript that addresses the points raised during the review process. Please submit your revised manuscript by Jan 06 2025 11:59PM. If you will need more time than this to complete your revisions, please reply to this message or contact the journal office at plosone@plos.org . Please include the following items when submitting your revised manuscript:

We look forward to receiving your revised manuscript.

Kind regards,

Mojtaba Kordrostami, Ph.D.

Academic Editor

PLOS ONE

Journal Requirements:

Reviewers' comments:

Reviewer's Responses to Questions

**Comments to the Author**

1. Is the manuscript technically sound, and do the data support the conclusions?

Reviewer #1: Yes

Reviewer #2: Yes

Reviewer #3: No

2. Has the statistical analysis been performed appropriately and rigorously? 

Reviewer #1: No

Reviewer #2: Yes

Reviewer #3: No

3. Have the authors made all data underlying the findings in their manuscript fully available?

Reviewer #1: Yes

Reviewer #2: Yes

Reviewer #3: Yes

4. Is the manuscript presented in an intelligible fashion and written in standard English?

Reviewer #1: Yes

Reviewer #2: Yes

Reviewer #3: Yes

5. Review Comments to the Author

Reviewer #1: The manuscript gives a good impact on the effect of various organic and inorganic fertilizers on the valuable agarwood seedlings. Valuable data was collected and presented in tables and figures, however, it lacks some statistical significance tests. It is recommended to use and apply some statistical analysis to check the the difference between treatments. e.g.

Table 1 and Table 2 should be supported horizontally each row with one way ANOVA to check the difference between the treatment groups, and vertically with either repeated measure ANOVA and/or Pearson's correlation. the change in seedling height with time should be presented at least by (r) Pearson's correlation coefficient and two-tailed significance test. Moreover, the overall table could be tested with two way repeated measure ANOVA (Factor1= treatment, Factor2= Time)

In addition, some minor corrections in the manuscript, including punctuations and other typing errors, should be revised.

Figure 1. caption line 181, the word Treat. would be expressed as "treatment." would be better

The figure legend isnot clear; if possible, clarify.

Line 106 in the materials and Methods part, "The average seedling height was 17.60±0.33..." please clarify which was used with the average (±SD) or average (±SE)

Reviewer #2: YES, manuscript technically sound, and do the data support the conclusions. Additon the authors provide the findings in their manuscript. Author applied statistical analysis which parallel finding conclusion. More language uses is understandable to reader.

Reviewer #3: There are some concerns about present manuscript as follow: the aquired data is belonged to one year study and so is not statistically more reliable. Moreover, There is noy any other external factor which studied beside fertilizer tretment.

6. PLOS authors have the option to publish the peer review history of their article (what does this mean? ). If published, this will include your full peer review and any attached files.

**Do you want your identity to be public for this peer review?** For information about this choice, including consent withdrawal, please see our Privacy Policy .

Reviewer #1: **Yes: ** Abdelghafar M. Abu-Elsaoud

Reviewer #2: **Yes: ** ARIF ALI

Reviewer #3: No

---

## [Author Response · Author response to Decision Letter 0]

15 Dec 2024

Reviewer 1

1.The manuscript gives a good impact on the effect of various organic and inorganic fertilizers on the valuable agarwood seedlings. Valuable data was collected and presented in tables and figures, however, it lacks some statistical significance tests. It is recommended to use and apply some statistical analysis to check the the difference between treatments. e.g.Table 1 and Table 2 should be supported horizontally each row with one way ANOVA to check the difference between the treatment groups, and vertically with either repeated measure ANOVA and/or Pearson's correlation. the change in seedling height with time should be presented at least by (r) Pearson's correlation coefficient and two-tailed significance test. Moreover, the overall table could be tested with two way repeated measure ANOVA (Factor1= treatment, Factor2= Time)

Re: Thank you for your patience in reviewing the manuscript and for your valuable comments. According to what you said, we have applied some statistical analysis to check the differences between the treatments, and at the same time added multiple comparisons between the treatments and the main effect of the months on the growth amount of seedling height and ground diameter, so as to make the statistical analysis more suitable and strict. Thank you again for your valuable advice.

2.In addition, some minor corrections in the manuscript, including punctuations and other typing errors, should be revised.

RE: Thank you for raising our spelling and punctuation questions. We have reviewed the full text thoroughly and made changes.

3.Figure 1. caption line 181, the word Treat. would be expressed as "treatment." would be better.The figure legend isnot clear; if possible, clarify.

RE: Thank you very much for examining the manuscript and asking this question. We have changed the "Treat" in Figure 1 and Figure 2 to "treatment." At the same time, we have modified the expression of the legend to make it clearer. Thank you again for your valuable advice.

4.Line 106 in the materials and Methods part, "The average seedling height was 17.60±0.33..." please clarify which was used with the average (±SD) or average (±SE)

RE: Thank you for your careful examination of the manuscript.We have corrected the sentence to:The average seedling height was 17.60 ± 0.33 cm (average ± standard deviation) and the average ground diameter was 4.95 ± 0.29 mm (average ± standard deviation).

Reviewer 2

YES, manuscript technically sound, and do the data support the conclusions. Additon the authors provide the findings in their manuscript. Author applied statistical analysis which parallel finding conclusion. More language uses is understandable to reader.

RE: Thank you so much for sparing your precious time from your busy schedule to review and comment on our research. Your professional and meticulous review has made us recognize the value of the article and has given us greater confidence in our research direction. Your recognition is an immense encouragement to us, and we will hold onto this encouragement to keep moving forward on the academic path and continue conducting relevant research. Sincere thanks to you once again!

Reviewer 3

There are some concerns about present manuscript as follow: the aquired data is belonged to one year study and so is not statistically more reliable. Moreover, There is noy any other external factor which studied beside fertilizer tretment.

RE: Thank you for your careful examination of the manuscript.

The design of our trial cycle was founded on some comparable literature that has been reported, and the experimental outcomes demonstrated similar results or trends to them. For instance, some researchers selected leaves on the 67th day after fertilization for the determination of physiological indicators (Kuanysh K， Ji D D， Kuanysh K， Hui W B， Li M L. Effects of Five Foliar Fertilizers on the Growth and Physiological Characteristics of Dahongpao Prickly Ash. Journal of Northwest Forestry University.2022;37 (4).135-142. DOI:10.3969/j.issn.1001-7461.2022.04.18), and some others measured indicators on the 80th day after fertilization(HE Jinjin, DENG Tiantian, LIU Qianyu, et al. Effects of different fertilization treatments on growth and physiological characteristics of Ardisia japonica[J]. Ecological Science, 2023, 42(4): 92–97). This indicates that our results are also compelling. However, you mentioned that due to the short test duration, the obtained results have certain limitations and are insufficient to represent the long-term fertilization effect. In the future, the test period will be prolonged, and further research will be conducted to make the test results more practical.

Regarding other factors that were not considered, we mainly utilized the pot control experiment to explore the influence of fertilizer on its growth. Other factors could almost be disregarded in this experiment, and we will also carry out the relevant research you proposed in the subsequent experiment.

Thank you again for your advice and we will be more careful with our trial design in future studies.

---

## [Decision Letter · Decision Letter 1]

25 Feb 2025

Effects of Organic-inorganic Complex Fertilizer  on the Growth and Physiological Characteristics of 'Qi-Nan' Agarwood from Aquilaria sinensis (Lour.)

PONE-D-24-45249R1

Dear Dr. Penglian,

We’re pleased to inform you that your manuscript has been judged scientifically suitable for publication and will be formally accepted for publication once it meets all outstanding technical requirements.

Kind regards,

Mojtaba Kordrostami, Ph.D.

Academic Editor

PLOS ONE

Additional Editor Comments (optional):

Reviewers' comments:

Reviewer's Responses to Questions

**Comments to the Author**

1. If the authors have adequately addressed your comments raised in a previous round of review and you feel that this manuscript is now acceptable for publication, you may indicate that here to bypass the “Comments to the Author” section, enter your conflict of interest statement in the “Confidential to Editor” section, and submit your "Accept" recommendation.

Reviewer #2: All comments have been addressed

2. Is the manuscript technically sound, and do the data support the conclusions?

Reviewer #2: Yes

3. Has the statistical analysis been performed appropriately and rigorously? 

Reviewer #2: Yes

4. Have the authors made all data underlying the findings in their manuscript fully available?

Reviewer #2: Yes

5. Is the manuscript presented in an intelligible fashion and written in standard English?

Reviewer #2: Yes

6. Review Comments to the Author

Reviewer #2: All the above sections include the English language; statistically, literature is written clearly and correctly. I would recommend proceeding further.

7. PLOS authors have the option to publish the peer review history of their article (what does this mean? ). If published, this will include your full peer review and any attached files.

**Do you want your identity to be public for this peer review?** For information about this choice, including consent withdrawal, please see our Privacy Policy .

Reviewer #2: No

---

## [Editor Report · Acceptance letter]

PONE-D-24-45249R1

PLOS ONE

Dear Dr. Penglian,

I'm pleased to inform you that your manuscript has been deemed suitable for publication in PLOS ONE. Congratulations! Your manuscript is now being handed over to our production team.

Kind regards,

on behalf of

Dr. Mojtaba Kordrostami

Academic Editor

PLOS ONE